# Synthesis of 3,4-Dihydropyrimidin(thio)one Containing Scaffold: Biginelli-like Reactions

**DOI:** 10.3390/ph15080948

**Published:** 2022-07-30

**Authors:** Francisco Sánchez-Sancho, Marcos Escolano, Daniel Gaviña, Aurelio G. Csáky, María Sánchez-Roselló, Santiago Díaz-Oltra, Carlos del Pozo

**Affiliations:** 1Instituto de Química Médica, CSIC, C/Juan de la Cierva 3, 28006 Madrid, Spain; francisco.sanchez@csic.es; 2Departamento de Química Orgánica, University of Valencia, Avda Vicente Andrés Estellés s/n, 46100 Valencia, Spain; marcos.escolano@uv.es (M.E.); daniel.gavina@uv.es (D.G.); maria.sanchez-rosello@uv.es (M.S.-R.); 3Instituto Pluridisciplinar, Universidad Complutense, Campus de Excelencia Internacional Moncloa, Paseo de Juan XXIII, 1, 28040 Madrid, Spain; csaky@ucm.es

**Keywords:** 3,4-dihydropyrimidinones, Biginelli, multicomponent reactions, privileged structures, biological activity

## Abstract

The interest in 3,4-dihydropyrimidine-2(1*H*)-(thio)ones is increasing every day, mainly due to their paramount biological relevance. The Biginelli reaction is the classical approach to reaching these scaffolds, although the product diversity suffers from some limitations. In order to overcome these restrictions, two main approaches have been devised. The first one involves the modification of the conventional components of the Biginelli reaction and the second one refers to the postmodification of the Biginelli products. Both strategies have been extensively revised in this manuscript. Regarding the first one, initially, the modification of one of the components was covered. Although examples of modifications of the three of them were described, by far the modification of the keto ester counterpart was the most popular approach, and a wide variety of different enolizable carbonylic compounds were used; moreover, changes in two or the three components were also described, broadening the substitution of the final dihydropyrimidines. Together with these modifications, the use of Biginelli adducts as a starting point for further modification was also a very useful strategy to decorate the final heterocyclic structure.

## 1. Introduction

The recurrent presence of some structural fragments in biologically important compounds and drugs with diverse biological activities was used by Evans to introduce the term “privileged structures” in 1988 [1] (later updated by Patchett and Nargund) [2]; these scaffolds are able to interact with more than one receptor or enzyme, playing a relevant role as a starting point in the drug discovery process. Among the fragments labelled as “privileged structures”, 3,4-dihydropyrimidinones (DHPMs) and their derivatives occupy a prominent place; these cores are of immense biological importance; play an important role as essential building blocks in the synthesis of DNA and RNA, and subtle changes in their structure provide a wide range of biological activities such as anti-inflammatory, anti-HIV, anti-tubercular, antifungal, anticancer, antibacterial, antifilarial, antihyperglycemic, antihypertensive, analgesic, anticonvulsant, antioxidant, anti-TRPA1 or anti-SARS, among others [3,4,5,6,7,8,9,10]; it is not surprising that the popularity of DHPMs lasts until present days, and the development of new methodologies to access these structural motifs is always of high interest [11,12,13,14].

Emblematic examples of DHPM-derivatives are monastrol and its derivatives enastron and piperastrol [15]; these compounds inhibit Kinesin-5, a protein involved in the regulation and function of mitosis, and are considered promising targets in cancer chemotherapy (Figure 1A–C). 5-Fluorouracil [16], due to its analogy with uracil, inhibits DNA formation by irreversible union with thimidilate synthase enzyme, inducing cell death (Figure 1D). After the discovery of the antitumoral properties of 5-fluorouracil, the incorporation of fluorinated moieties into organic molecules became a fundamental strategy in medicinal chemistry. (*S*)-L-771688 is the first α_1a_-adrenoceptor selective antagonist to be tested in the clinic for the treatment of benign prostatic hyperplasia (Figure 1E) [17]. (*R*)-SQ 32926, considered a close structural analog of the therapeutically widely used calcium channel blockers of the 1,4-dihydropyridine type (e.g., nifedipine) displayed interesting hypertensive properties (Figure 1F) [18]. Nitractin is highly effective against trachoma virus and also shows some antibacterial activity (Figure 1G) [19]. Idoxuridine, initially developed as an anticancer drug, became an antiviral agent used for the topical treatment of herpes simplex keratitis (Figure 1H) [20]. Batzelladine A belongs to a family of polycyclic guanidine alkaloids that inhibit the binding of HIVgp-120-CD4 (Figure 1I) [21]. Finally, emivirine was developed as an agent for the treatment of HIV as a non-nucleoside reverse transcriptase inhibitor (Figure 1J) [22].

The classical approach to accessing DHPM derivatives is the Biginelli reaction. Initially, it involved the acid-catalyzed cyclocondensation of a urea or thiourea **1**, aromatic aldehyde **2**, and a beta-keto ester **3** (Figure 1). First described in 1893 [23,24], it remained almost unexplored until the 1980s when the power of multicomponent reactions was recognized as a useful tool for medicinal chemists. The convergent character, operational simplicity, easily accessible and molecular diverse starting materials as well as the atom economy of this transformation make the Biginelli reaction one of the most important multicomponent processes in drug discovery, that continuously attracts research interests due to the occurrence of DHPMs in biologically active products and drugs. On account of this research activity, a wide variety of methodologies have been devised for the racemic and asymmetric Biginelli reaction, together with a big number of reaction conditions that involved homogeneous and heterogeneous catalysis, the use of ionic liquids as the solvent, immobilized catalysts in solid supports or the use of microwave irradiation [25,26,27,28,29,30,31,32,33,34,35,36]. Additionally, a big number of variants of this multicomponent reaction that gain access to novel DHPMs by modification of the original fragments in a multicomponent or step-wise manner have been described.

The Biginelli reaction was thoroughly reviewed from both, synthetic and pharmacological points of view, covering all the literature advancements in this field. In contrast, Biginelli-like reactions, where one or more components of the multicomponent process were modified, were only treated by Jie-Ping Wan in 2010 [10]. Therefore, the present review summarizes other routes to access DHMPs beyond the Biginelli reaction updating the new methodologies developed to this end. The classification of the review is depicted below: Section 2: Modification of the urea counterpart. Section 3: Modification of the aldehyde counterpart. Section 4: Modification of the ketoester counterpart—Section 4.1: Diketones as ketoester component; Section 4.2: β−Keto amides as ketoester component; Section 4.3: β−Keto acids as ketoester component; Section 4.4: Ketones as keto ester component; Section 4.5: Other different substrates as keto ester component. Section 5: Modifications of two components. Section 6: Modifications of all components—Alternative routes to dihydropyrimidinones. Section 7: Structure diversification of 3,4-dihydropyrimidin-2-(1*H*)-(thio)one derivatives.

## 2. Modification of the Urea Counterpart

The first important change of the urea counterpart in a Biginelli-type reaction was the Atwal modification [37,38,39]. In this approach, *O*,*S*-substituted isoureas **5** together with a preformed unsaturated carbonyl compound **6** were condensed in a basic medium (Figure 2). Through this methodology, it was possible to improve the efficiency of the Biginelli synthesis, especially with aliphatic and aromatic aldehydes slightly hindered by *ortho*-substituents. The unsaturated carbonyl compound **6** was obtained via Knoevenagel condensation from the corresponding β-keto esters **3** and aldehydes **2** in a separated synthesis.

Using this approach, Rovnyak et al. prepared uniquely designed dihydropyrimidines **8**–**10** starting from conveniently functionalized α-benzylidene β-keto esters **6**; these derivatives were synthesized in order to establish structural and conformational determinants in calcium channel modulation (Figure 3) [40].

However, preformation of the α-benzylidene β-keto esters was not mandatory in all cases and the classical three-component reaction could be performed directly in a basic medium using *O*-methyl isourea **5**, ethyl acetoacetate **3** and substituted benzaldehydes **2** to form DHPMs **11** (Figure 4); these compounds were derivatized through the selective reaction with phenacyl bromides **12** at N3, to obtain derivatives **13** that showed good antihypertensive, anti-inflammatory, and analgesic activity, as well as low ulcerogenic activity. The N3 selectivity could be a consequence of the richer electron density of N3 compared to N1 [41].

Some other variations of the urea component in Biginelli-like reactions include the use of guanidine to obtain 2-amino-1,4-dihydropyrimidines, however, the direct three-component Biginelli reaction with guanidine is useful only with benzoyl acetates and aryl aldehydes, and fails to give useful yields using acetoacetates. A more general Biginelli-based method for preparing 2-imino-5-carboxy-3,4-dihydropyrimidines **18** was developed by Nilsson and Overman in 2006 [42]. Two alternatives are shown in Figure 5, starting from pyrazole carboxamidine **14** in a four-step sequence, or starting from the triazone-protected guanidine **15** in a two-step sequence. Both alternatives utilize acetoacetates **3** and are compatible with aliphatic aldehydes.

Ultrasound irradiation has also been used to promote the direct three-component Biginelli reaction with guanidine hydrochloride, acetoacetates and aromatic aldehydes [43].

At the beginning of the XXI century, the Biginelli reaction was extended by replacing the urea component with 5-amino-1,2,4-triazoles. Theoretically, four possible compounds could be obtained taking into account two regioisomers with or without dehydration. In fact, 3-alkylthio-5-amino-1,2,4-triazoles **19** gave two different dihydrotriazolo-pyrimidines **20** and **21** (Figure 6), the selectivity strongly depending on the substitution of the reactants [44].

*N*-Substituted ureas and thioureas have also been used in Biginelli-like reactions [45,46,47]. Inspired by the mechanisms of biocatalysts, in 2016, the Saá research group developed the concept of noncovalent organocatalysis by means of networks of cooperative hydrogen bonds, utilizing arylideneureas **22** in the asymmetric reaction with ethyl acetoacetate **3** (Figure 7). Arylideneureas can act as donor-acceptor in hydrogen bonds, thus they are capable of assembling with the chiral catalyst **I** and at the same time activate the nucleophile, rendering final DHPMs **4** with excellent enantioselectivities [48].

The use of selenourea **23** as the starting material is one of the most efficient methods for the synthesis of selenium-containing heterocycles [49], and it has been used to prepare selenoxopyrimidines **24** by means of a one-pot multicomponent reaction with ethyl acetoacetate **3** and aromatic aldehydes **2** in acidic medium (Figure 8). The synthesized compounds were shown to possess a significant antimicrobial and anticancer activity in vitro [50].

Finally, *N*H-free sulfonimidamides **25** have also been used as the urea component in Biginelli-type multicomponent reactions (MCRs), to provide 2,3-dihydro-1,2,6-thiadiazine 1-oxides **26**, generally in high yields (Figure 9). As the sulphur in sulfonimidamides is stereogenic the reaction produces two diastereoisomers with variable selectivities. The couplings are performed in a planetary ball mill (PM) under solvent-free mechanochemical conditions, catalysed by acetic acid or ytterbium triflate [51].

## 3. Modification of the Aldehyde Counterpart

The original Biginelli reaction was described with aromatic aldehydes. Although the reaction with aliphatic aldehydes is less efficient, their participation in this MCR is now widespread and, in this section, the use of aliphatic aldehydes will not be considered as a modification of the aldehyde component.

The first reported example of a modification at the aldehyde building block in the Biginelli reaction was the use of acylals **27** [52]; this masked carbonyl functionality, together with ethyl acetoacetate **3** and urea or thiourea **1** were employed to give dihydropyrimidinones **4** under acid catalysis. The reaction was catalyzed by 12-tungstophosphoric acid (PW), 12-molybdophosphoric acid (PMo) or zinc chloride and performed in a one-pot procedure under solvent-free conditions (Figure 10).

The best conditions to prepare the DHPMs were achieved when 10 mol%, 20 mol%, and 80 mol% of PW, PMo, and ZnCl_2_ were used, respectively; these optimum conditions were applied to a series of substituted aromatic acylals **27** observing that with electron-donating containing substituents the reaction proceeded faster and gave high yields in short reaction times.

Stucchi et al. described the use of *N*-substituted isatins **28** as carbonyl substrates in an asymmetric, Brønsted acid catalyzed Biginelli-like reaction [53]. The use of BINOL-derived phosphoric acid catalyst **II** allowed the authors to obtain enantioenriched spiro(indoline-pyrimidine)-diones derivatives **29** in moderate-to-good yields (Figure 11).

The halogen substitution at the aryl ring had little effect on both yield and *ee*. Likewise, methyl and benzyl acetoacetates provided similar results leading to good yields and moderate *ee*’s in the final products; however, the *N*-Me isatin gave a better result than the corresponding *N*-benzyl, *N*-*p*-nitrobenzyl, and *N*-*p*-methoxybenzyl ones in terms of yield (93% to up to 63%), although suffering a drop in *ee* (50% to up to 80%). Surprisingly, neither thiourea in place of urea nor various linear or cyclic β-diketones instead of alkyl acetoacetates gave good results. With thiourea, no reaction occurred, whereas, with β-diketones, a complex mixture of products was obtained.

Recently, A. A. Malik and coworkers described the synthesis of dihydropyrimidones **4** via sequential Kornblum oxidation/Biginelli reaction [54]. The method involves the in-situ generation of benzaldehydes **2** from benzyl halides **30**, under catalyst-free conditions, which were subsequently converted into dihydropyrimidones **4** in a one-pot manner under microwave (MW) irradiation (Figure 12).

The key step for this transformation was the oxidation of benzyl halide to aldehyde using Kornblum oxidation conditions [55]. The best synthetic results for this oxidation were obtained when the reaction was performed under microwave irradiation at 80 °C using DMSO as the solvent and in the absence of a catalyst. Under these conditions, the tandem one-pot synthesis of dihydropyrimidones was achieved by reacting benzyl bromide (1.0 mmol) in the presence of urea (1.0 mmol) and ethyl acetoacetate (1.2 mmol) in 1.5 mmol of DMSO.

The scope of the reaction was evaluated using different substituted benzyl halides. In general, both electron-withdrawing and electron-releasing substituents attached to the benzyl substrate reacted successfully and afforded the desired products in good yields; however, benzyl substrates with electron-withdrawing substituents needed more time for completion with lower yields compared to benzyl halides with electron-releasing groups.

The product purification through aqueous recrystallization avoids the use of large quantities of volatile and toxic organic solvents, which makes the method environmentally and nature-friendly.

## 4. Modification of the Keto Ester Counterpart

### 4.1. Diketones as the Keto Ester Component

1,3-diketone compounds have been extensively used as substrates for the Biginelli reaction as the enolizable component. We will review below the most representative examples.

Shaabani and coworkers described the use of ammonium chloride as a catalyst in a one-pot Biginelli condensation reaction of aldehydes **2**, 1,3-dicarbonyl compounds **31**, and urea or thiourea **1** under solvent-free conditions [56]. The best results were obtained with a 0.5:1:1:1.5 ratio of ammonium chloride, aldehyde, 1,3-dicarbonyl compound, and urea or thiourea. Particularly, the use of acetylacetone as the dicarbonyl compound afforded the corresponding 3,4-dihydropyrimidin-2-(1*H*)-ones **32** with good yields, under these conditions (Figure 13).

Montmorillonite has also been used as an efficient environmentally friendly catalyst for the synthesis of 3,4-dihydropirimidine-2(1*H*)-ones under one-pot three-component Biginelli reaction under solvent-free conditions. The use of acetylacetone as the 1,3-dicarbonyl counterpart afforded the corresponding product in good yield [57]. 

More recently, the use of acetylacetone as substrate in a one-pot synthesis of 3,4-dihydropyrimidin-2-(1*H*)-ones and 3,4-dihydropyrimidin-2-(1*H*)-thiones catalyzed by Bi(NO_3_)_3_·5H_2_O or ZrCl_4_, respectively, under solvent-free conditions, was reported by Matias et al. [58,59]. The in vitro antiproliferative activity and QSAR studies of the synthesized compounds were also described.

Gartner et al. published the synthesis and biological evaluation of several analogs of monastrol. The different analogs were synthesized as racemic mixtures by using the Biginelli reaction [60]. The most potent analogs, with enhanced inhibition of Mitotic Kinesin Eg5 compared to monastrol, were those named by the authors as enastron **34**, dimethylenastron **35**, and enastrol **36** (Figure 14). The irradiation of a mixture of a cyclic diketone **33**, 3-hydroxybenzaldehyde, and thiourea, in a domestic microwave oven, together with the use of polyphosphate ester (PPE) afforded enastron **34** and dimethylenastron **35** with moderate yields. Enastrol **36** was synthesized as a 3:1 diastereomeric mixture from enastron **34** by selective Luche reduction [61] of the 5-carbonyl function (Figure 14).

The most potent compound, dimethylenastron **35**, is up to more than 100-times more potent than monastrol, both in vitro and with arresting mitosis of cultured cells; these novel inhibitors have the potential to be interesting anticancer drug candidates.

Independently, Kidwai and coworkers described the synthesis of 4-aryl-7,7-dimethyl-1,2,3,4,5,6,7,8-octahydroquinazoline-2-one/thione-5-one derivatives **37** using a Biginelli reaction in the absence of solvent and catalyst and under microwave irradiation, employing neat reaction conditions (Figure 15) [62]. The use of aromatic aldehydes **2**, urea/thiourea **1**, and 5,5-dimethyl- 1,3-cyclohexanedione **33** (dimedone) as the 1,3-dicarbonyl compound, led to the corresponding quinazoline derivatives **37** in good yields. The synthesized compounds were screened for their in vitro antibacterial activity against standard strains of *Staphylococcus aureus*, *Escherichia coli* and *Pseudomonas aeruginosa*.

Later, Abnous and coworkers reported the synthesis of six Biginelli compounds through one-step Biginelli reaction of dimedone **33** with three imidazole aldehydes, and urea or thiourea using chlorotrimethylsilane (TMSCl) as a catalyst. The products were evaluated for their in vitro cytotoxicities and their inhibitory effects on ATPase activity of kinesin [63]. 

Following a similar process, Niralwad et al. described the microwave-assisted one-pot synthesis of octahydroquinazolinone derivatives in high yields using dimedone, urea/thiourea, and appropriate aromatic aldehydes under ammonium metavanadate (NH_4_VO_3_) as a catalyst under solvent-free conditions [64]. Likewise, Badadhe and coworkers reported the use of 10 mol% of thiamine hydrochloride (VB1) as an efficient catalyst affording good to excellent yields [65]. Additionally, Shah and coworkers published the synthesis of some new octahydro-quinazolinone derivatives using zinc triflate (Zn(OTf)_2_) as a catalyst, in refluxing ethanol, in high yield [66].

More recently, 1,3-cyclohexanedione or dimedone has been employed by Silva and coworkers as substrates for the Biginelli synthesis of 3,4-dihydropyrimidin-2(1*H*)-one or thione (DHPMs) derivatives catalyzed by two novel coordination polymers (CPs) under solvent-free conditions and heterogeneous catalysis. The reaction conducted under continuous flow conditions afforded very promising results toward a scale-up of the reaction [67]. 

Bariwald et al. reported the use of benzoylacetone as the 1,3-dicarbonyl counterpart in the Biginelli reaction together with urea/thiourea and several substituted benzaldehydes in ethanol with a catalytic amount of conc. HCl. The newly synthesized molecules were screened for their anti-proliferative activity [68]. 

Another example of a Biginelli reaction with PPE catalysis employing 1,3-diketones is the synthesis of a series of trifluoromethylated hexahydropyrimidine and tetrahydropyrimidine derivatives **39** was described by Agbaje et al. (Figure 16) [69]; these fluorinated compounds were evaluated for their in vitro cytotoxic activities in a colon cancer cell line (COLO 320 HSR).

Interestingly, Azizian and coworkers presented the first synthesis of novel derivatives of bis(dihydropyrimidinone)benzenes **41** using chlorotrimethylsilane (TMSCl) as the catalyst through the reaction of terephthalic aldehyde, 1,3-dicarbonyl compounds **40** and (thio)urea or guanidine **1** under microwave irradiation conditions (Figure 17) [70]; this Biginelli condensation method provided products containing two different dihydropyrimidinone units and allowed their obtaining in with high yields (>85%) and in short reaction times (4–6 min).

The cytotoxic activities of these compounds were evaluated on five different human cancer cell lines. Their cytotoxic study indicated that they possessed a weak-to-moderate activity.

The same Biginelli reaction, catalyzed by TMSCl, in dimethylformamide as the solvent, at room temperature, without the use of microwave irradiation, was employed by Zhu et al. [71], for the synthesis of several dihydropyrimidine derivatives. Three of them derived from the use of 1,3-diketones as the 1,3-dicarbonyl counterpart in the Biginelli reaction. The dihydropyrimidine derivatives were subsequently coupled with homocamptothecin to obtain novel conjugates (hCPT-DHPM); these conjugates were effective cytotoxic agents that showed also superior Topo I inhibition activity than hCPT itself.

The synthesis of tricyclic 3,4-dihydropyrimidine-2-thione derivatives **43** was described by Gijsen et al. [72], via a Biginelli three-component reaction between indane-1,3-dione **42**, thiourea, and several substituted benzaldehydes **2** (Figure 18). Subsequent derivatization led also to some *N*-methylated compounds. 

All products were tested on both the human and rat TRPA1 channel. For two of the most interesting compounds, the racemates were separated into the enantiomers, showing that only the dextrorotary enantiomers were active. The absolute configuration of the active enantiomer was determined to be 4*R*.

Recently, the use of indane-1,3-dione has also been described for the synthesis of 3-substituted 5-phenylindeno-thiazolopyrimidinone derivatives. The Biginelli reaction proceeded under solvent-free conditions, using Poly(4-vinylpyridinium)hydrogen sulfate as the catalyst [73]. All the synthesized molecules were investigated for their antimicrobial potency against different microbes.

Lal and coworkers described the synthesis of curcumin derivatives using a one-pot cyclocondensation of curcumin (**44**) with substituted aromatic aldehydes **2** and urea/thiourea/guanidine **1** in the presence of different catalysts [74,75,76]. The use of chitosamine hydrochloride as a biodegradable and nontoxic catalyst, under solvent-free conditions and using microwave irradiation, allowed the authors to synthesize curcumin 3,4-dihydropyrimidinones/thiones/imines **45** in excellent yields. All compounds were evaluated for their antioxidant and anti-inflammatory activity (Figure 19) [76]. 

The synthesis of oxygen-bridged monastrol analogs **47** was reported by Cheng et al. using a Biginelli reaction of substituted salicylaldehydes **2**, acetylacetone (**46**), and urea or thiourea **1** with NaHSO_4_ as the catalyst under microwave irradiation and solvent-free conditions in a short time and with good yields (Figure 20) [77]. 

Similarly, the synthesis of this kind of oxygen-bridged monastrol analogs by a Biginelli reaction was recently reported using a natural acidic medium of *Averrhoa bilimbi* extract (ABE) as an eco-friendly and economically cheap, non-toxic acidic catalytic media [78]. The advantages of this process are excellent yields of the obtained products, versatility in handling substrates, reuse of the catalyst, use of no hazardous organic solvents, and minimization of waste or side products.

Ramos and coworkers described the synthesis, characterization, and application of a new ion-tagged recyclable iron catalyst to the Biginelli reaction [79]. The synthesis of dihydropyrimidine derivatives was performed by using MAI·Fe_2_Cl_7_ (5 mol%), different aromatic and aliphatic aldehydes, urea or thiourea, and 1,3-dicarbonyl compounds at 80 °C for 2 h. In particular, the use of acetylacetone or dimedone as 1,3-dicarbonyl compounds led to the corresponding products in good yields; these products are potent Eg5 inhibitors and have potent activity against MCF-7 and MDA-MB-231 cells [80]. 

The use of dimedone or indane-1,3-dione as the 1,3 dicarbonyl compound in the Biginelli reaction was reported by Siddiqui et al. for the synthesis of novel bis-3,4-dihydropyrimidin-2(1*H*)-one derivatives **48** in excellent yields using perchloric acid-modified poly(ethylene)glycol 6000 (PEG-HClO_4_) as a biodegradable and reusable catalyst at ambient temperature under solvent-free condition at 70 °C (Figure 21) [81]. 

Recently, Davanagere and coworkers have reported a modified procedure for the synthesis of 1,3-bis(carboxymethyl)imidazolium chloride [BCMIM][Cl], a metal-free ionic catalyst, and its application in the one-pot multicomponent Biginelli reactions to dihydropyrimidine-2(1*H*)-ones/thiones **49** in solvent-free conditions (Figure 22) [82]. 

A recent example of the use of acetylacetone as the 1,3-dicarbonyl compound in the Biginelli reaction is the synthesis of 5-acetyl-6-methyl-4-(1,3-diphenyl-1*H*-pyrazol-4-yl)-3,4-dihydropyrmidin- 2(1*H*)-thione, that was achieved in high yield by one-pot three-component synthesis using CaCl_2_ in refluxing EtOH [83]; this compound was used by the authors as starting material to synthesize a new series of 5-pyrazolyl; isoxazolyl; pyrimidinyl derivatives and also fused isoxazolo[5,4-*d*]pyrimidine and pyrazolo[3,4-*d*]pyrimidine; these compounds were evaluated for their antibacterial, antifungal and anti-inflammatory activity.

### 4.2. Keto Amides as the Keto Ester Component

#### 4.2.1. Barbituric Acid Derivatives

Shaabani and coworkers [84,85] described the efficient synthesis of spiro-fused heterocycles using conventional heating [85] or microwave irradiation [84] under solvent-free conditions. The microwave-assisted one-pot method involves the heating of a mixture of Meldrum’s acid or barbituric acid derivatives **50** (instead of open chain cyclic β-dicarbonyl compounds), urea **1** and an aromatic aldehyde **2** in the presence of a protic acid catalyst to give a series of novel heterobicyclic compounds **51** in good yields and in a stereoselective manner (Figure 23). The best catalyst for the formation of the spiro-fused compounds was found to be acetic acid or NaHSO_4_.

Similarly, Mohammadi and coworkers reported the synthesis of spiropyrimidinethiones/spiropyrimidinones-barbituric acid derivatives [86]. The one-pot reaction of barbituric acid, different benzaldehydes and urea or thiourea in the presence of a nanoporous acid catalyst of SBA-Pr-SO_3_H, under solvent-free conditions, afforded novel heterobicyclic compounds in good yields. The spiro compounds were tested for their urease inhibitory activity against Jack bean urease [87].

The use of thiobarbituric acid derivatives **52** was later reported by Dabholkar et al. [87] for the Biginelli reaction with aromatic aldehydes **2** and urea or thiourea **1** using a catalytic amount of concentrated HCl in refluxing ethanol (Figure 24). Representative samples of the synthesized compounds **53** were screened for their anti-microbial activity.

The synthesis of a series of 5-indolylpyrimido[4,5-*d*]pyrimidinones **55** was reported by Gupta and coworkers [88] by means of a multi-component reaction of 3-formylindoles **54**, thiobarbituric acid/barbituric acid **52** and thiourea/urea (**1**) in dry media (Figure 25). The reaction proceeded under conventional heating, microwave irradiation (MW), or grinding together neat reactants to give the titled compounds good-to-high yields. Representative compounds were also evaluated for their antimicrobial activity.

The synthesis of substituted pyrimido[4,5-d]pyrimidinones **57** using a Biginelli-like reaction was reported by Rimaz et al. [89]; this transformation proceeds through a three-component tandem annulation of arylglyoxalmonohydrates **56** with 1,3-dimethylbarbituric acid **52** and thiourea **1** in the presence of catalytic amounts of 1,4-diazabicyclo[2.2.2]octane (DABCO) or L-proline. Later on, the same authors described the one-pot regioselective and chemoselective synthesis of the above-mentioned derivatives in water using two green catalytic systems (ZrOCl_2_·8H_2_O and DABCO); the desired products were obtained in good to excellent yields (Figure 26) [90]. 

#### 4.2.2. Beta-Ketoamides and Beta-Ketosulfonamides

A series of conformationally flexible and restricted dimers of monastrol were described by Kamal and coworkers using a one-pot Biginelli multicomponent reaction [91]. The β-keto amide intermediate **58**, a derivative prepared from L-proline, was used as the 1,3-dicarbonyl compound in the Biginelli condensation with dibenzaldehydes **59** and thiourea **1** to obtain the asymmetric dimers **60** (Figure 27); these dimers were evaluated for cytotoxic potency against selected human cancer cell lines and some of the compounds exhibited more cytotoxic potency than the parent monastrol. In addition, the DNA binding ability and antimicrobial activities of these compounds were also evaluated, but with little success.

The synthesis of diarylpyrazole-ligated dihydropyrimidines possessing a lipophilic carbamoyl group **63** was reported by Yadlapalli et al. [92]. The use of acetoacetanilide derivatives **61** as the 1,3-dicarbonyl compound in the Biginelli reaction with 1,3-diaryl-1*H*-pyrazole-4-carbaldehydes **62** and urea/thiourea **1**, afforded the corresponding dihydropyrimidine derivatives **63** with good-to-high yield (Figure 28); these novel compounds showed moderate anticancer activity against MCF-7 breast cancer cell lines as well as good to excellent antitubercular activity against MTB H37Rv.

A series of novel 1,2,3,4-tetrahydropyrimidine derivatives were synthesized, in moderate to good yields, by Elumalai et al. [93] by reacting *N*-(3,5-dichloro-2-ethoxy-6-fluoropyridin-4-yl)-3-oxobutanamide **64**, urea or thiourea **1** and aromatic aldehydes **2** in the presence of a catalytic amount of *p*-toluen sulfonic acid (*p*-TsOH) (Figure 29). The newly synthesized compounds **65** were evaluated for their antimycobacterial activity against *Mycobacterium tuberculosis*.

Later on, the same authors published the use of acetazolamide derived ketoamide **66** as a substrate for the Biginelli condensation with urea or thiourea **1** and aromatic aldehydes **2** under microwave irradiation (Figure 30) [94]. The synthesized compounds **67** were evaluated for in vitro antimicrobial and cytotoxicity against *Bacillus subtilis*, *Escherichia coli* and *Vero* cells.

More recently, these authors published the synthesis, antimicrobial activity, and in vitro cytotoxicity of novel sulphanilamide condensed 1,2,3,4-tetrahydropyrimidines [95]. 

Chikhale et al. described the synthesis of novel derivatives of benzothiazolyl pyrimidine-5-carboxamides **69** which were synthesised by an acid-catalyzed one-pot three-component reaction of benzothiazolyl oxobutanamide **68**, substituted aryl aldehydes **2** and thiourea **1** (Figure 31). The resulting products **69** were evaluated for their antitubercular activity to determine MIC against *Mycobacterium tuberculosis* (H37Rv) [96].

Ramachandran and coworkers [97] reported the syntheses of dihydropyrimidinones using the solvent-free grindstone chemistry method: a mixture of an aromatic aldehyde, *N*-phenylacetoacetamide, urea/thiourea, cupric chloride, and a few drops of concentrated HCl was ground together to give the desired dihydropyrimidinones. The products were studied for their antibacterial activity.

Likewise, Gein and coworkers described the synthesis of 1,2,3,6-tetrahydro-pyrimidine-5-carboxamides by reacting arylacetoacetamides with aromatic aldehydes and urea under solvent-free conditions at 120–150 °C for 5–7 min [98]. Good yields of the target compounds were obtained and the study of their antimicrobial activity was reported.

Recently, the parallel synthesis of new Biginelli 1,4-dihydropyrimidines **71** was reported by Faizan et al. [99]. The desired compounds were synthesized via parallel synthesis by multicomponent-cyclisation reaction between aliphatic, aryl, heteroaryl aldehydes, *o*-methyl acetoacetanilide **70**, and excess of urea or thiourea **1** in absolute ethanol and using *p*-toluen sulfonic acid as catalyst (Figure 32). 

Good yields were obtained and evaluation of anticancer activity and structure-activity relationships via 3D QSAR studies were carried out on the products.

### 4.3. Keto Acids as the Keto Ester Component

The use of β-keto carboxylic acids as substrates in the Biginelli reaction is scarce, given that a typical β-keto carboxylic acid should undergo spontaneous decarboxylation to give carbon dioxide and a ketone under the standard acidic reaction conditions; however, oxalacetic acid **72** does not undergo decomposition, presumably because the enol form is stabilized by resonance for both acids. Thus, replacing alkyl acetoacetate with oxalacetic acid **72** as a substrate for the Biginelli reaction led to the formation of 5-unsubstituted 3,4-dihydropyrimidin-2-(thio)-ones **73** due to in situ decarboxylation after cyclization [100]. The major drawback of this reaction is the long reaction time (12 h); however, it can be conducted expeditiously with good yield and applied to a variety of reagents under microwave irradiation (Figure 33) [101].

One special case of functionalized keto carboxylic acids is aromatic γ or δ-keto acids **74** (Figure 34); these are used as enolizable ketones in Biginelli-like reactions because aryl alkanoic acids are the most studied class of non-steroidal anti-inflammatory drugs (NSAIDs), such as diclofenac sodium, naproxen, ibuprofen, etc. Therefore, pyrimidine derivatives **75**, with acetic or propanoic acid moiety at the fifth position, were synthesized to study their anti-inflammatory activity in vivo through the base-catalyzed condensation of aromatic γ or δ-keto acids **74**, thiourea **1**, and the appropriate aldehyde **2**. Propanoic acid derivatives **75** (*n = 2*) showed significant anti-inflammatory activity, due to their improved lipophilicity compare to acetic acid derivatives **75** (*n* = 1) [102,103].

### 4.4. Ketones as the Keto Ester Component

In contrast to numerous protocols available for Biginelli reactions, Biginelli-like reactions using enolizable ketones instead of β-keto esters have been less explored [104]. In 2004, Holla and coworkers synthesized 4,6-diaryl-3,4-dihydropyrimidin-2(1*H*)-thiones **78** in a two-step protocol. Firstly, condensation of 2,4-dichloro-5-fluoroacetophenone **76** with benzaldehydes **2** under Claisen–Schmidt reaction conditions led to the corresponding chalcones **77**. In a second step, these chalcones reacted with thiourea **1** in the presence of ethanolic potassium hydroxide to render final DHPMs **78** in good yields (Figure 35) [105].

This type of reaction can be performed directly by the classical three-component one-pot synthesis with different systems. For instance, acetophenone **79** reacted with substituted benzaldehydes **2** and urea **1** in a microwave-assisted Biginelli-like reaction in a short and concise manner employing ZnI_2_ as a catalyst under solvent-free conditions to afford DHPMs **80** (Figure 36) [106]. The same reaction can be catalyzed by MnO_2_/CNT(carbon nanotubes) nanocomposites with very good activity, recovery, and reusability of the catalyst [107].

In 2020, Desai et al. reported a simple methodology for the synthesis of pyrimidinthione derivatives **82** via the condensation of substituted acetophenones **79**, a pyrazol-4-carbaldehyde **81**, thiourea **1** and sulfamic acid as the catalyst, in order to test their antimicrobial activity. TMSCl is believed to promote aromatization of the intermediates DHPMs (Figure 37) [108].

The first asymmetric catalytic version of Biginelli-like reactions using enolizable ketones as substrates was described by Li et al. in 2009 [104]; they found that BINOL-derived chiral organocatalyst **III** was able to catalyze the reaction of cyclic and acyclic aliphatic ketones **83** with aromatic aldehydes **2** and *N*-benzyl thiourea **1**, yielding DHMPs **84** in excellent enantioselectivities (Figure 38). Aromatic ketones like 1-*p*-tolylethanone gave only moderate enantioselectivity (61% *ee*), and enolizable aliphatic aldehydes underwent a self-Biginelli-like reaction excluding the ketone from the reaction.

The chiral derivative of 1,2-benzenedisulfonimide **IV** was found to be an efficient Brønsted acid catalyst to perform the standard Biginelli reaction of β-keto esters enantioselectively, with very high yields and excellent enantiomeric excesses. Surprisingly, when using acetophenones **79** as enolizable ketones instead of β-keto esters, two consecutive cyclizations occurred leading to the *meso* form of adducts **85** in high yields (Figure 39); it seems that the nature of the acid catalyst and the absence of steric hindrances are decisive in leading the reaction towards this type of adducts [109].

In the case of using cyclohexanone **86** as the enolizable ketone, the low cost and facile to prepare TADDOL-derived chiral phosphoric acid **V** (obtained from natural tartaric acid) could be used to catalyze the Biginelli-like reaction with aromatic aldehydes **2** and *N*-benzyl thiourea **1**. The resulting enantioselectivity depends on the aldehyde substitution pattern (Figure 40) [110].

Unlike bigger cycloalkanones, cyclopentanone **88** has been described to furnish aryliden fused pyrimidinones **89** (Figure 41) through double α-reaction of the ketone, instead of the classical Biginelli-like product through single α-reaction [111]; this transformation has been performed using different catalytic systems, most of them based on acidic ionic liquids (ILs). In the example shown in Figure 41 (conditions **a**), Rahman et al. employed the Brønsted acid catalyst **VI** under microwave irradiation and solvent-free conditions to produce heterobicyclic dihydropyrimidinone derivatives **89**. The ionic liquid used as the catalyst could be reused at least six times without any noticeable decrease in catalytic activity. Attempts to expand the Biginelli-type reaction to condensations of cyclohexanone and/or aliphatic aldehydes lead to multiple unidentified products [112]. Later on, the group of Professor Lu compared the efficiency of different Brønsted acidic ionic liquid catalysts in this transformation and concluded that the eco-friendly catalyst **VII** gave the best results and could be reused at least seven times without significant loss of catalytic activity (Figure 41, conditions b) [113].

More examples of this transformation include the use of ILs immobilized in zeolites as catalysts [114], or the use of AlCl_3_ in poly(ethylene)glycol (PEG) as a green and reusable solvent [115]. 

Aromatic cyclic ketones such as 1-indanone **90** (Figure 42) have also been used in Biginelli-type condensation with substituted benzaldehydes **2** and thiourea **1** to afford 4-aryl-1,3,4,5-tetrahydro-2*H*-indeno[1,2-*d*]pyrimidine-2-thiones **91** under microwave irradiation; these thiones were converted into their *S*-alkylated/arylated derivatives in order to evaluate their antibacterial activities [116].

Similarly, fused DHPMs **93** were obtained by the condensation of 6-methoxy-1-tetralone **92**, aromatic aldehydes **2**, and urea or thiourea **1**, in the presence of acidic IL **VIII** as the catalyst under solvent-free conditions in excellent yields (Figure 43) [117].

The utilization of enolizable aldehydes instead of ketones is scarce; however, in 2013 Qu et al. reported a highly chemo- and regio-selective tandem reaction of alkyl aldehydes **94**, arylaldehydes **2** and mono-substituted urea **1**, to give highly diverse 6-unsubstituted DHPMs **95** in reasonable yields under mild reaction conditions. The authors developed two different methods to carry out the reaction. Thus, method A (Figure 44) involved the use of molecular iodine as the catalyst, whereas method B represented the first catalytic enantioselective version of this reaction, by using chiral spirocyclic SPINOL-phosphoric acid **IX**. Although the resulting enantioselectivities (*ee* values) were from low-to-moderate (Figure 44) [118].

A different family of 6-unsubstituted DHPMs **97** was prepared in 2016 by Bhat et al. using enaminones **96** as a surrogate of the enolizable carbonylic compound, with aromatic aldehydes **2** and urea or thiourea **1** in acetic acid (Figure 45); these DHPMs were evaluated for antitumor activity against cancer stem cells in vitro, and one of them (R^1^ = MeO, R^2^ = 4-EtOC_6_H_4_, X=O) demonstrated a remarkable antitumor effect in colon cancer xenografts in mice [119].

Another family of compounds with anticancer activity are the mono- and di(1,4-disubstituted 1,2,3-triazole)-DHPM hybrids **100** and **101**, respectively (Figure 46). On the one hand, the monotriazole-DHPM hybrids **100** were synthesized by a one-pot multicomponent reaction involving a copper(I)-catalyzed alkyne-azide cycloaddition (CuAAC) and a Biginelly-like reaction, starting from phenylacetylene **98**, 1-azidopropan-2-one **99**, urea **1** and aromatic aldehydes **2**. On the other hand, a multistep sequence of reactions that included bromination, azidation, and a CuAAC afforded the ditriazole-DHPM hybrids **101** [120].

To conclude with the variants of ketones utilized as enolizable carbonylic compounds in Biginelli-like reactions, the condensation of phenylacetone **102**, aromatic aldehydes **2**, and thiourea **1** to give 4-aryl-5-phenyl-4-methyl substituted DHPMs **103**, in the presence of potassium carbonate nanoparticles (NPs) to promote the reaction, was reported very recently; the antimicrobial activity of the synthesized compounds was also evaluated (Figure 47) [121].

### 4.5. Other Different Substrates as the Keto Ester Component

The first example of this section was reported by Yadav et al. employing unprotected aldoses **106** as bio renewable aldehyde component and changing the keto ester counterpart to the mercaptoacetylating active methylene building block, 2-methyl-2-phenyl-1,3-oxathiolan-5-one **105**, in turn prepared from acetophenone **79** and 2-mercaptoacetic acid **104** [122]. The MCR takes place by heating under solvent-free microwave irradiation at 90 °C the aldose **106**, urea or thiourea **1**, the mercaptoacetylating agent **105**, and the nanoclay montmorillonite K-10. In this manner, the thiosugar-annulated DHPMs **110** were obtained in good yields and excellent diastereoselectivities. Initially, the acetylating agent **105** underwent Knoevenagel-type condensation with the aldose to render intermediate **107**. (Thio)urea **1** was then added in a Michael type addition to render **108**, which eliminates acetophenone intermolecularly to yield the key intermediate **109**. Intramolecular condensation of the sugar moiety and the thiol provided final products **110** (Figure 48).

The same authors described the synthesis of perhydropyrimidines changing the keto ester moiety for 2-phenyl-1,3-oxazol-5-one **111**, and using again unprotected aldoses **106** as renewable aldehyde counterparts [123]. Again, the MCR took place under solvent-free microwave irradiation, in the presence of cerium sulphate as the catalyst, rendering iminosugar annulated perhydropyrimidines **112** and **113** in good yields and excellent diastereoselectivies. The outcome of the process is similar to the one shown in Figure 48 and, after the conjugated addition of the (thio)urea, the second nitrogen opened the isoxazolone ring with a final intramolecular condensation of the first urea nitrogen with a hydroxyl group of the sugar (Figure 49).

One year later, the same authors reported an interesting variation of the previously developed reactions [124]. In this case, the aldehyde counterpart was an aromatic aldehyde (**2**), and the MCR reaction with the (thio)urea **1** and 2-methyl-2-phenyl-1,3-oxathiolan-5-one **111** or 2-phenyl-1,3-oxazol-5-one **105** as the β-keto ester substitute, was performed in a chiral ionic liquid [(Pro)_2_SO_4_]. Under those conditions, the corresponding polyfunctionalized perhydropyrimidines **115** and **116** were obtained in excellent yields and enantioselectivies (Figure 50).

Myrboh et al. described the synthesis of a series of fused pyrimidine derivatives by the three-component reaction of an aryl aldehyde **2**, urea **1** (or guanidine) in the presence of 1,3-dimethyl-dihydropyrimidine-2,4-dione **117** as the keto ester counterpart by heating the mixture in dioxane with coated alumina (KF-alumina) for 3–5 h [125]. Following an analogous mechanism to the Biginelli reaction, the process gave rise to a family of 7-pyrimido[4,5-*d*]pyrimidin-2-ones **118** in good yields. The process was further extended the synthesis to pyrrolo[2,3-*d*]pyrimidines and pyrido[2,3-*d*]pyrimidines **120** (n = 1 and 0, respectively) employing substituted acetophenones **79** instead of the aldehyde component and 1-methyl-1*H*-pyrrol-2(3*H*)-one or 1,1-methylpiperidin-2-one **119** (n = 1 and 0, respectively) as the keto ester counterpart. In this manner, and by heating the reaction mixture in ethanol at 80 °C, good yields of the fused pyrimidines **120** were achieved (Figure 51).

Naliapara and coworkers found that etidronic acid is an efficient catalyst for the MCR of aromatic aldehydes **2**, urea **1**, and 1-(2-hydroxyphenyl)-2-nitroethanone **121** [126]. Etidronic acid is less acidic than other phosphoric acids such as polyphosphoric acid and does not affect sensitive aldehydes. The reaction took place in THF under microwave irradiation to provide excellent yields of the nitro-DHPMs **122** (Figure 52).

A Biginelli-type three-component reaction was developed by Shah et al. for the synthesis of a new family of pyrimidine derivatives [127]; they used malononitrile **123** as the keto ester component, thiourea **1**, and aromatic aldehydes **2** in refluxing methanol. Resulting DMHPs **124** were treated with dimethyl sulfate and oxidized to the corresponding pyrimidines **125** in good yields; these compounds display interesting antibacterial activities (Figure 53).

Wan and coworkers reported a Biginelli-like reaction initiated by a secondary amine, changing the keto ester component by a propiolate **126** [128]. The secondary amine would react with the propiolate rendering a beta-enaminone that would act as the keto ester moiety through an enamine-type addition. The heating of the three-component mixture with the base in DMF at 90 °C provided the corresponding DHMPs **127** unsubstituted at position 6 (Figure 54); it is noteworthy that this substitution is difficult to access with regular Biginelli conditions.

In 2015 Sekar et al. described the synthesis of novel Biginelli scaffolds using an eco-friendly method that involved the use of the halogen-free ionic liquid *N*-methyl-2-pyrrolidonium hydrogen sulfate [(HNMP)+(HSO_4_)] [129]. The use of ionic liquids is advantageous over conventional solvents, due to the shortening of reaction times, recyclability, and from a green chemistry point of view. The multicomponent reaction was performed with aromatic aldehydes, urea, and changing the keto ester component, either for 3-methyl-1-phenyl-5 (4*H*)-pyrazolone or naturally occurring 2-hydoxy-4-naphthoquinone (Lawsone) as a source of active methylene groups. The reaction takes place both under conventional heating at 80 °C or with ultrasonic irradiation at rt to render the corresponding fused DHMPs in good yields. Those derivatives were further applied as dispersed dyes on polyester and nylon fibers.

Shah et al. described the use of pyrazolones as the keto ester component in Biginelli-like reactions [130]. 1-Phenothiazine pyrazolone **128** is a rare case of enolizable ketone counterpart used in Biginelli-like reaction to produce pyrazolopyrimidinethiols **129** with a phenothiazine heterocycle embedded in the structure; these compounds exhibited anti-tubercular activity (Figure 55).

Another modified Biginelli reaction was performed by Selvi et al. [131]. In this case, the keto ester component was 6-methyl-4-hyrdoxyquinolin-2(1*H*)-one **130**, and the MCR with aromatic aldehydes **2** and phenyl urea **1** provided, under microwave irradiation, 1,4-dihydropyrimido[5,4-c]quinolones **131** in excellent yields (Figure 56).

Gill and coworkers recently developed a green methodology for the synthesis of pharmacology promising pyrimidine-2,4-diones [132]. Based on the use of room-temperature ionic liquids (RTILs), the authors employed diisopropyl ethyl ammonium acetate (DIPEAc) as the solvent of the MCR of (thio)ureas **1**, aldehydes **2**, and ethyl cyanoacetate **132** as the keto ester counterpart. After 45 min at room temperature, excellent yields of the DHMP-derivatives **133** were obtained, even with aliphatic aldehydes (Figure 57). The ionic liquid was recycled without loss of efficacy. The final products displayed interesting in vitro antibacterial and antifungal properties.

Yu, Yin and coworkers reported a transition-metal and Brønsted acid co-catalyzed MCR of (thio)urea **1**, aromatic aldehydes **2**, and alkynols **134** as a Biginelli-like reaction for the synthesis of spirofuran-hydropyrimidinones **135** [133]. Alkynols **134** are versatile organic synthons; under metal catalysis, they could undergo intramolecular hydroalkoxylation to render unusual enolizable carbonyl equivalents. The author combined this process with the presence of a Brønsted acid co-catalyst that would promote the condensation of the urea and the aldehyde. When PdCl_2_ as a metal catalyst and trifluoroacetic acid as a Brønsted acid were combined in this reaction, the corresponding Biginelli reaction took place in excellent yields and complete diastereoselectivities, to afford a variety of spirofuran-hydropyrimidinone compounds **135** (Figure 58); this is one of the few reports that deal with the synthesis of spirocyclic DHMP-derivatives.

Bálint et al. described the synthesis of DHPM-containing phosphonates at the 5 position by using beta-ketophosphonates **136** as the keto ester component [134]. The MCR reaction took place under solvent-free microwave irradiation to render the corresponding Biginelli adducts **137** in good yields, even with aliphatic aldehydes (Figure 59). The use of ketophosphonates in the Biginelli reaction was also reported with Zn(OTf)_2_ in refluxing toluene [135], or using acetic acid [136,137] or *p*-toluene sulfonic acid [138] as catalysts.

Finally, Kidway et al. reported a combination of the Biginelli and Hantzsch reactions [139]; they used thiobarbituric acids **52** as the keto ester component and they were combined in a MCR with aromatic aldehydes **2** and ammonium acetate **138**. After heating under microwave irradiation in the presence of Al_2_O_3_ as the catalyst, excellent yields of hybrid compounds **139** were obtained (Figure 60).

## 5. Modification of Two Components

Among the different strategies employed to date for synthesizing Biginelly-type 3,4-dihydropyrimidin(thio)ones, the aldehyde counterpart is usually fixed whereas the (thio)urea and the β-keto ester components are modified; however, in the first example of this section, the urea component is fixed and both, the aldehyde and β-keto ester functionalities were substituted by resin-bounded γ-ketosulfones **144** (Figure 61); these polymer-supported compounds were prepared in three steps from a polystyrene resin functionalized with sodium sulfonate **140**: (i) sulfinate *S*-alkylation to give derivatives **141**, (ii) sulfone anion alkylation with an epoxide **142**, and (iii) oxidation of the γ-hydroxyl sulfones **143**. Finally, reaction with (thio)urea **1** in basic medium afforded DHPMs **145** in variable overall yields [140].

Multicomponent condensation reactions of 5-aminopyrazoles **146** with cyclic 1,3-diketones **33** and aromatic aldehydes **2** can lead to the formation of several different tricyclic reaction products due to the presence of at least three non-equivalent nucleophilic reaction centers in the aminopyrazole building block **146** (N1, C4, and NH_2_); however, Chebanov et al. demonstrated in 2008 that the reaction can be kinetically controlled to produce pyrazoloquinazolinones **147** under ultrasound irradiation (Figure 62) [141].

The first simple and efficient approach towards the one-step synthesis of 2-amino-5-cyano-6-hydroxy-4-aryl pyrimidines **149** by three-component condensation of aromatic aldehydes **2**, ethyl cyanoacetate **132**, and guanidine hydrochloride **148** in alkaline ethanol, was developed by Deshmukh et al. in 2009. The synthesized compounds were evaluated for their antibacterial activity and some of them showed excellent activity against Gram-positive and Gram-negative bacteria (Figure 63) [142].

An efficient and convenient approach for the synthesis of [1,2,4]triazolo/benzimidazolo quinazolinones **151** and **153**, respectively, was reported by Heravi et al. in 2010 [143]. The method is based on the condensation of 3-amino-1,2,4-triazole **150** or 2-amino benzimidazole **152** as nitrogen sources, with dimedone **33** and different aldehydes **2** in the presence of sulfamic acid as catalyst (Figure 64).

In 2011, Ryabukhin et al. reported a systematic investigation on the use of aminoheterocycles as synthons for combinatorial Biginelli reactions, to generate combinatorial libraries comprising more than 2000 compounds of high structural and functional diversity. A representative set of 89 compounds was described (Figure 65) [144].

A one-pot three-component cyclocondensation of isatoic anhydride **154**, NH_4_OAc, and aromatic/heteroaromatic aldehydes **2**, was efficiently catalyzed by montmorillonite K-10 to produce the corresponding 2-substituted-2,3-dihydroquinazolin-4(1*H*)-ones **155** in very good yields. The group of 2-(2-chloroquinolin-3-yl)-2,3-dihydroquinazolin-4(1*H*)-ones were screened for their antitumor activity (Figure 66) [145].

If benzamidine hydrochloride **156** is employed instead of urea, 2,6-diaryl-4-(3*H*)-pyrimidinones **157** and 2,6-diaryl-4-aminopyrimidines **158** can be obtained through the reaction with aromatic aldehydes **2** and ethyl cianoacetate **132**, or malonitrile **123**, respectively (Figure 67); this eco-friendly synthesis was performed in water under microwave irradiation. Some of these derivatives showed significant analgesic activity in mice [146].

Gupta et al. reported in 2014 the use of green chemical techniques, namely solvent-free microwave irradiation and grindstone technology, to make 2-amino-5-cyano-6-hydroxy-4-aryl pyrimidines **160** starting from substituted 3-formylindoles **159** as aromatic aldehydes; they also evaluated their antimicrobial activities (Figure 68) [147].

The synthesis of benzimidazoloquinazolinones **161** can be promoted by heterogeneous Fe_3_O_4_@chitosan as a superparamagnetic nanocatalyst, under mild reaction conditions, using 2-aminobenzimidazole or 2-aminobenzothiazole **152** as nitrogen source, dimedone **33** and aromatic aldehydes **2** (Figure 69) [148].

Thiadiazoloquinazolinones **163** can also be prepared from diketones **33**, aldehydes **2**, and thiadiazoloamines **162** as nitrogen sources, by an on-water microwave-assisted reaction catalyzed by *p*-toluen sulfonic acid. Notably, not only do aromatic aldehydes work well in this process, but formaldehyde or acetaldehyde work as well (Figure 70) [149]; this reaction with aromatic aldehydes has also been performed in water-ethanol mixture with tetrabutylammonium hydrogen sulfate as the catalyst, and some of the thiadiazoloquinazolinones obtained showed potent antioxidant activity [150].

The condensation reaction between benzylamine and pyrazolecarbaldehydes **164** with isatoic anhydride **154** in basic medium afforded pyrazol-4-yl-2,3-dihydroquinazolin-4(1*H*)-ones **165** in excellent yields. The optical and electrochemical properties of the compounds were studied, together with their anticancer activity (Figure 71) [151].

Magnetic nanoparticles (MNPs) can be used in heterogeneous catalysis to facilitate the recovery of the catalyst, as seen with the Fe_3_O_4_@chitosan nanoparticles to catalyze the formation of benzimidazolo quinazolinones in Figure 69. An improvement of this methodology was reported by Kamali and Shirini in 2017 which consisted of using Fe_3_O_4_@SiO_2_–ZrCl_2_-MNPs to enable the solvent-free synthesis of benzimidazolo quinazolinones **161** (Figure 72) [152].

The application of microwave irradiation and scandium triflate as catalyst permitted, as well, the solvent-free synthesis of benzimidazolo quinazolinones **161**, employing 2-aminobenzimidazole or 2-aminobenzothiazole **152** as nitrogen source, diketones **33** and aromatic aldehydes **2** (Figure 73) [153].

An efficient microwave-promoted three-component synthesis of thiazolo[3,2-*a*]pyrimidines **167** catalyzed by SiO_2_–ZnBr_2_, employing diisopropylethylamine as a base, was developed starting from thiazol-2-amines **162**, 2-(4-nitrophenyl)acetonitrile **166**, and aromatic aldehydes **2** (Figure 74) [154]. In addition, the antimicrobial activity of these compounds was evaluated.

Pyrimidine derivatives **169** and **170** were prepared from aromatic aldehydes **2**, ethyl cyanoacetate **132**, and guanyl hydrazone derivatives **168**, in the presence of piperidine as a catalyst (Figure 75). Under these conditions, some of the 3,4-dihydropyrimidines intermediates are directly oxidized to aromatic pyrimidines **170**. The starting guanylhydrazones **168** were prepared by the Knoevenagel condensation of the respective aldehydes or ketones with aminoguanidine hydrochloride (not depicted). The synthesized compounds were evaluated for their antitumoral activity [155].

An alternative to forming benzimidazoloquinazolinones **161** without the use of metallic catalysts was the utilization of thiamine hydrochloride, also known as vitamin B_1_ (**X**), as an organocatalyst in an aqueous medium. The process makes use of 2-aminobenzothiazole **152** as a nitrogen source, dimedone **33**, and aromatic aldehydes **2** (Figure 76) [156].

Unlike aldehydes, ketones are not reactive enough to engage in the classical Biginelli reaction acting as electrophiles and they always react as a nucleophile via enolization; however, very recently, Nishimura et al. prepared the key intermediate **171** by using TiCl_4_ and pyridine in a Knoevenagel-type condensation (Figure 77) [157]. Thus, the cyclocondensation reaction of this compound **171** with *O*-methylisourea hemisulfate salt **5** gave a tautomeric mixture of dihydropyrimidines **172** and **173**; this mixture was hydrolyzed to produce 4,4-disubstituted 3,4-dihydropyrimidin-2(1*H*)-ones **174** in high yields; these products had been inaccessible and hitherto unavailable for medicinal chemistry and were assessed for their antiproliferative effect on a human promyelocytic leukemia cell line, HL-60.

## 6. Modification of All Components—Alternative Routes to Dihydropyrimidinones

An enantioselective synthesis of DHMPs in a three-step sequence, starting from beta-dicarbonyl compounds **175** and α-amido sulfones **176** as acyl imine precursors was developed by Schaus et al. [158,159]. The reaction took place in the presence of cinchonine as a catalyst in basic media to render beta-amino ketones **177** with excellent enantioselectivities. Alloc protecting group release in the presence of an isocyanate rendered the corresponding ureas **178** that cyclized into the desired DHPMs **179** by heating in ethanol under microwave irradiation (Figure 78); this methodology was further applied by the same author to the enantioselective synthesis of SNAP-7941, a chiral DHPM Inhibitor of MCH1-R (melanin-concentrating hormone receptor antagonist) [160].

In 2009, Fustero and coworkers developed a new synthetic strategy to access fluorinated DHMPs [161]; it involved a one-pot process with ethyl-3-butenoate **180**, fluorinated nitriles **181**, and iso(thio)cyanates **182**. The reaction was initiated by the formation of the ester enolate by the addition of LDA (lithium diisopropyl amide) at −78 °C. The ambident nature of this enolate allows the addition to the alfa- or gamma positions. After some optimization, the authors found that with a very slow addition of the fluorinated nitrile **181** to the enolate, the gamma addition took place preferentially, rendering lineal enamino ester **183**; this intermediate reacted in a one-pot manner with heterocumulenes **182**, by initial nucleophilic addition followed by an intramolecular aza-Michael reaction (IMAMR), to render fluorinated DHPMs **184** in moderate yields (Figure 79).

Perumal et al. reported the synthesis of pyridopyrimidine-2-thiones in a four-component MCR with *N*-substituted 4-piperidones **185**, two equivalents of an aromatic aldehyde **2** and thiourea **1** [162]. Although two components are the same as in the classical Biginelli reaction, this example has been included in this section because the process needed two equivalents of the aromatic aldehyde. After 2 min with sodium ethoxide in a solvent-free reaction, excellent yields of pyridopyrimidine-2-thiones **189** were obtained. Therefore, initially, the double condensation with the piperidone with two equivalents of the aldehyde occurred, rendering bis-enone **186**, which underwent a conjugated addition with the thiourea. Finally, condensation of the other nitrogen with the carbonyl group followed by tautomerization accounts for the formation of bicyclic DHMPs **189** (Figure 80).

Pal and coworkers developed a MCR with isatoic anhydride **154**, 2-formyl benzoic acid **190**, and amines **191** to access dihydroisoindolo [2,1-*a*]quinazoline-5,11-dione derivatives **192** [163]; this Biginelli-like reaction was performed under the catalysis of montmorillonite K10 in refluxing ethanol, providing DHPM-derivatives **192** in excellent yields (Figure 81). Those compounds were found to be potent inhibitors of TNF-α (tumor necrosis factor-alpha), key cytokine mediators in the inflammatory response. The same authors developed a greener route to access DHMP-derivatives **192** using cyclodextrins as catalysts in aqueous media under microwave irradiation [164]. After heating the MCR in a sealed vial at 120 °C under microwave irradiation for 10 min, comparable yields to the previous protocol were obtained (Figure 81). 

El-Gohary et al. described the use of 2-amino-1,3,4-thiadiazole-5-sulfonamide **193** for the synthesis of a wide variety of fused DHPM-derivatives (Figure 82) [165]. Cyclocondensation of amine **193** with benzylidene derivatives of diethyl malonate **194** in DMF afforded ethyl 5,6-dihydro-7-oxo-5-(substituted)phenyl-2-sulfamoyl-[1,3,4]thiadiazolo[3,2-*a*]pyrimidine-6-carboxylates **195**. With ethyl benzylidenecyanoacetates **196**, compounds **197** analogous to **195** containing a cyano group were obtained. 5-Amino-7-oxo-6-[4-(substituted)phenyldiazenyl]-5H,7H-[1,3,4]thiadiazolo[3,2-*a*]pyrimidine-2-sulfonamide derivatives **199** were obtained by reaction with ethyl 2-cyano-2-[4-(substituted)phenyldiazenyl]acetates **198** in refluxing AcOH. Reaction with benzylidene tetralones **200** in refluxing propylene glycol rendered benzo[h][1,3,4]thiadiazolo[2,3-*b*]quinazoline derivatives **201**. With ethyl 2-(ethoxymethylenecyano)acetate **202** or 2-(ethoxymethylene) malononitrile **204** in refluxing glacial acetic acid, compounds **203** and **205** were obtained, respectively. The reaction of **193** with malononitrile **123** in absolute ethanol and in the presence of a catalytic amount of triethylamine afforded 5-amino-7-imino-7H-[1,3,4]thiadiazolo[3,2-*a*]pyrimidine-2-sulfonamide **206**. Finally, treatment of **193** with chloro propiopyl chloride **207** in refluxing AcOH afforded 6,7-dihydro-7-oxo-5H-[1,3,4]thiadiazolo[3,2-*a*]pyrimidine-2-sulfonamide **208** (Figure 82). All these compounds were screened in several biological assays; it was found that they display interesting antibacterial, antifungal and anticancer properties.

Sośnicki and coworkers developed an alternative route to the Biginelli reaction to access DHPM-derivatives [166]; it involved the use of pyrimidine-2(1*H*) thiones **211**, prepared from 2-aryl-3-(dimethyl-amino)allylidene(dimethyl)ammonium perchlorates **210** (easily available from aryl acetic acids **209**) by reaction with thioureas **1** in the presence of MeONa. With this pyrimidine thiones **211** in hand, the addition of aryl lithiums afforded 5,6-diaryl-DHPMs **212** in good yields (Figure 83); this methodology, complementary to the Biginelli reaction, was used also with other different organometallic reagents [167,168,169,170,171]. Resulting 4,5-diaryl compounds exhibited important inhibition of cell proliferation, especially for breast cancer, when compared to monastrol.

Ma and coworkers described the enantioselective synthesis of benzofused-DHPMs **215** by means of a highly enantioselective decarboxylative Mannich reaction between β-keto acids **214** and trifluoromethyl ketimines **213** [172]. The reaction took place in the presence of chiral sugar-derived thiourea **XI**, giving rise to 3,4-dihydroquinazolin-2(1*H*)-one derivatives in excellent yields and outstanding enantioselectivities (Figure 84). The process was further extended for the same authors to non-fluorinated ketimines [173].

Choudhury et al. described the synthesis of novel 5,6-disubstituted pyrrolo[2,3-*d*]pyrimidine-2,4-diones **219** and **220** via three-component reactions between 6-amino uracil derivatives **216**, aryl glioxal **217** and either thiols or malononitrile [174]. The reaction, which could be performed under conventional heating or microwave irradiation, in short reaction times and good yields to access these biologically interesting pyrrolo-pyrimidinedione derivatives (Figure 85).

Panahi and Khalafi-Nezhad et al. described an MCR of carbohydrates, barbituric acid, malononitrile or diethyl malonate and aromatic aldehydes catalyzed by TsOH [175]. Initially, they performed the reaction with sugars **221**, malononitrile, diethyl malonate, or ethyl-2-cyanoacetate (**223**) with barbituric acid **222**. After heating the mixture in ethanol at 50 °C for 12 h, excellent yields of pyrano[2,3-*d*]pyrimidine derivatives **224** were obtained (Figure 86). Additionally, the authors optimized a four-component reaction with glucosamine **225**, barbituric acid **222**, malononitrile **123** and aromatic aldehydes **2**. Again, by heating the mixture in ethanol under the presence of catalytic amounts of TsOH, excellent yields of pyrido[2,3-*d*]pyrimidine derivatives **226** were obtained (Figure 86). The synthetic compounds shown before revealed excellent antioxidant properties.

Shi and Zhou et al. reported an original synthesis of enantiomerically enriched DHMPs by asymmetric biomimetic transfer hydrogenation of pyrimidines catalyzed by a chiral phosphoric acid (CPA) [176]. The reaction took place from pyrimidine derivatives **227** and Hantzsch ester **228** in toluene at 40 °C with (*R*)-TRIP-PA **XII** as CPA for 24 h, rendering final Biginelli products **229** with excellent yields; it is noteworthy that the best results were obtained with aromatic substituents at R^1^ and R^2^ reaching enantioselectivities up to 99%, while a clear drop in efficiency occurred with aliphatic substituents (Figure 87).

Jian, Shi and coworkers described the synthesis of a new family of DHMPs containing a heteroatom at C4 by nucleophilic dearomatization of 2-hydroxy pyridines **230** [177]. The reaction took place under the presence of catalytic amounts of ZrCl_4_ with phosphite esters **231** as nucleophiles rendering DHMP-containing phosphonic esters **232** (Figure 88).

König reported an environmentally benign protocol for the synthesis of 5-unsubstituted dihydropyrimidinone-4-carboxylate derivatives from deep eutectic mixtures [178]. The process involved the solvent-free heating of β,γ-unsaturated ketoesters **233** in low melting l-(+)-tartaric acid–*N*,*N*-dimethylurea mixtures, to render DHPMs **234** in good yields (Figure 89).

Sondhi et al. described the preparation of several DHPM-derivatives by reaction of 4-isothiocyanato-4-methylpentan-2-one **236** with several amines and diamines **235** containing a wide variety of carbo- and heterocycles [179]. After the addition of the amine to the isothiocyanate at room temperature, the resulting thiourea cyclized with the carbonyl moiety by dehydration after heating in refluxing methanol (Figure 90). Those derivatives showed interesting anti-inflammatory properties.

## 7. Structure Diversification of 3,4-Dihydropyrimidin-2-(1*H*) (thio)one Derivatives

Due to the high biological relevance of DHPMs, the generation of new derivatives that contain this scaffold is very important in Medicinal Chemistry. One of the best ways to generate diversity in the structure of the DHPM core is the decoration of the Biginelli adducts. In 2005 Kappe and Dallinger reviewed the advances in this field [180], which are summarized in Figure 91. Modification at N1 is normally made by alkylation through treatment of DHMPs **4** with a base and alkyl halides since this proton is more acidic than the proton at N3; it was also described the introduction of phosphates using this strategy. Modification of C2 is probably the less usual one. With DHPM thiones as starting materials, it is possible to alkylate the sulfur atom. Alternatively, with Nickel-Raney, sulfur can be removed. The regioselective N3-acylation can be performed with Vilsmeier reagent or, alternatively, with acid chlorides or anhydrides. The protection with carbamates is troublesome and renders mixtures of both regioisomers. Substitution at position 4 is related to the starting aldehyde employed in the Biginelli reaction. Although the process is more consistent with aromatic aldehydes, several efficient methodologies have been devised with aliphatic or heteroaromatic aldehydes, or even ketones to render the corresponding spiro compound (see Section 3). Normally, modifications at C5 are related to the keto ester component. Usually, an ester group is used and, therefore, those modifications are related to ester chemistry, by conversion to carboxylic acids, acyl azides, or isocyanates, among others. Finally, since acetoacetates are usually involved in the Biginelli reaction, the usual substitution at C6 is a methyl group. Therefore, some strategies have been described to modify this methyl group, such as bromination and further substitution with a wide variety of nucleophiles (Figure 91).

With these precedents in mind, this section will cover advances from 2004. Additionally, the use of more complex strategies leading to polycyclic analogues of DHPMs was also reviewed from this point. On the other hand, chemical resolutions of racemic DHPMs were not considered in this review.

Kappe and coworkers performed the modification of Biginelli adducts bearing a methyl group at C6 [181]. Following a bromination/azidation sequence, they installed the azide functionality to carry out the click chemistry. The authors perform the bromination of substrates **4** with a modified macroporous Amberlyst A-26 polystyrene resin in perbrominated form **238**; this solid-supported brominating agent was used under flow conditions to perform the bromination of the methyl group at C6. Bromo derivatives **239** were transformed into the corresponding azides **240** by heating with sodium azide in DMF. Finally, the dipolar cycloaddition was accomplished by reaction with alkynes **241** under copper catalysis, rendering the final DHMPs **242** modified at C6 in good yields (Figure 92).

Singh et al. developed a route to DHMPs functionalized at the 6 position by metallation and addition of an electrophile [182]. DHMP adducts **4** bearing a methyl group at C6 were treated with LDA and the subsequent addition of an electrophile to render the desired DHMPs **243** (Figure 93). 

The same research group found that, when DHPMs were metallated at −78 °C, and the reaction quenched with an acid chloride, regioselective acylation at N3 occurred, rendering acyl DHPMs **244** in good yields (Figure 94) [183].

Evans and coworkers employed a chiral DHMP as starting material for the total synthesis of the tricyclic guanidine core of the natural product Batzelladine C [184]. The steps started with a rhodium-catalyzed allylic amination reaction of Biginelli adduct **245**. The generation of the bicyclic core of the compound was accomplished by means of a highly diastereoselective radical 1,4-addition. The tricyclic guanidine core was built using known methodologies to finally render Batzelladine D (Figure 95).

Mobinikhaledi et al. described the intramolecular Friedel-Crafts acylation reactions of DMPMs **4** bearing an ester group at position 6 [185]. The reaction took place in nitrobenzene at 90 °C in the presence of acetyl chloride and AlCl_3_ to render fused (thio)oxopyrimidine derivatives **250** in moderate yields (Figure 96).

Ashok and coworkers functionalized DHMPs with a thiazole ring, generating new bicyclic structures that have shown moderate to good growth inhibition of bacteria and fungi [186]. To this end, dihydropyrimidinthiones **4** were treated in an MCR with chloroacetic acid and the corresponding aryl aldehyde **2** in the presence of anhydrous sodium acetate in refluxing acetic acid. In this manner, the corresponding thiazolo[2,3-*b*]pyrimidin-3-(1*H*)-ones **251** were obtained in good yields (Figure 97).

Singh et al. found that is possible to metalate DHMP derivatives **4** with butyl lithium (3.5 equiv at −10 °C), to further react with dielectrophiles, rendering N1, C6-linked bicyclic DHPMs **252** [187]. The reaction proceeded in excellent yield for the formation of five-membered rings, while the formation of the corresponding six-membered ring took place in low yield (Figure 98); these bicyclic scaffolds constitute important structural features of several alkaloids.

The same authors described the use of *N*,*N*-diacyl DHPMs as acylating agents [188]. Biginelli adducts **4** were diacylated at N1 and N3 by metalation with butyl lithium followed by the addition of an acyl chloride. Diacylated DHPMs **253** were found to be excellent acyl transfer reagents to ammonia, primary and secondary amines, to render primary, secondary and tertiary amides **254**, **255**, and **256**, respectively in excellent yields (Figure 99). Additionally, the resulting Biginelli adduct **257** *N*-monoacylated could be recycled.

Shin et al. described the synthesis of 2-unsubstituted pyrimidine rings from Biginelli products, not easily accessible [189]. The oxidation of dihydropirimin-2(1*H*)-thiones with oxone on wet alumina or hydrogen peroxide in the presence of a catalytic amount of vanadyl sulfate provided dihydropyrimidines **258** that were further oxidized to 2-unsubstituted pyrimidines **259** in good yields with KMnO_4_. When the attempt of aromatization with KMnO_4_ was performed over starting DHMPs **4**, 2-hydroxypyrimidine products **260** were obtained in moderate yields (Figure 100).

Similar oxidation protocols of DHPMs could also be performed with *tert*-butyl hydroperoxide in the presence of a copper(II) catalyst [190] or stoichiometric (diacetoxyiodo) benzene [191].

Saidi et al. described the synthesis of thiadiazoloquinazolinone derivatives **263** by reaction with 3-amino-2-mercaptoquinazolinone **261** and dialkylacetylendicarboxylates **262** in refluxing DMF [192]. After the initial 1,4-addition of the thiol moiety, intermediate **264** could undergo the intramolecular aza-Michael reaction either via a 5-exo-trigonal mode (***a***) or via a 6-endo-trigonal mode (***b***) (Figure 101). Only products from via ***a*** were detected.

Gong and coworkers developed an asymmetric enantioselective version of the Biginelli reaction using chiral BINOL phosphoric acids as catalysts [104]; they employed this methodology for the synthesis of compound **4**, employed for the treatment of benign prostatic hyperplasia. Biginelli adduct **4** was obtained with 91% *ee*. The derivatization of the DHPM started with the sequence oxidation/bromination to obtain compound **265**. Methoxylation was performed with sodium methoxide to obtain compound **266** without loss of enantiomeric excess. The introduction of the lateral chain of the final drug was introduced using known methodologies (Figure 102).

Singh and coworkers developed a new methodology for the regioselective alkylation at N1 of DHPMs [193]. Biginelli adducts **4** were treated under phase transfer catalysis conditions with tetrabutyl ammonium hydrogen sulfate and 50% aqueous NaOH and several alkyl halides, to afford excellent yields of the N1-alkylated DHPMs **267** (Figure 103). Those compounds showed a minor calcium channel blocking activity when compared to nifedipine.

DHMPs are potential inhibitors of dihydrofolate reductase, which is an interesting target for the treatment of mycobacterial infections; however, dihydropyrimidines are not represented in the current clinical treatments of tuberculosis. Shah and coworkers found that a postmodification of Biginelli adducts provided potential compounds for the treatment of micobacteria [194]. To this end, DHMPs **4** containing a pyrazol moiety were treated with dimethyl sulfate (DMS) in basic media to afford *S*-methylated products **268** (Figure 104). Two of those compounds resulted in being more potent than isoniazide.

Rajanarendar et al. described the modification of isoxazolyl dihydropyrimidine-thione carboxylates **269** by condensation with isoxazole amine [47]. After heating the mixture in diphenyl ether at 200 °C, a new tricyclic scaffold **270** was formed in good yields. The biological evaluation of the products obtained showed that they exhibited good antibacterial and antifungal properties when compared to standard antibiotics (Figure 105).

The same research group described the transformation of Biginelli adducts in tetrahydropyrimidines [195]. The desulfuration of DHPM **4** took place under mild conditions with Ni-Raney, affording compound **271** with an iminic moiety suitable to react with nucleophiles. Thus, the addition of organolithium or Grignard derivatives proceeded in good yields to render the corresponding tetrahydropyridines **272** (Figure 106); those compounds displayed cytostatic properties.

Again, Singh et al. described the transformation of Biginelli adducts into pyrimidines [196]. The initial step was the oxidation of DHPMs **4** with pyridinium chlorochromate (PCC) [197]. The next step was the treatment of compounds **273** with POCl_3_, to render chloropyrimidine derivatives **274** that were finally converted into final pyrimidines **275** by exposure to the corresponding amines or alcohols (Figure 107). In vitro evaluation of those derivatives showed that they display inhibitory activity of *Mycobacterium tuberculosis* and they are modulators of cytostatic activity.

When this protocol was performed with 4-aminoquinolines as final amines, the corresponding pyrimidines showed interesting in vitro anti-plasmodian properties [198].

Singh et al. also described the modification of Biginelli adducts at N3 [199]. Starting from DHPMs **4**, protected at N1 (since is the most reactive nitrogen), the treatment with POCl_3_ at 105 °C afforded compounds **276**, with the phosphorus oxychloride group at N3. Those derivatives are very reactive and were used without purification. Thus, reaction with ammonia gas yielded diaminophosphinyl DHPMs **277** in good yields. Reaction with primary amines or ethanol afforded the corresponding addition products **278** in moderate yields. Finally, the use of diamines or aminoalcohols rendered the corresponding cyclic DHPMs **279** in good yields (Figure 108); those derivatives were subjected to calcium chanel binding studies albeit they were less active than nifedipine.

Aly and coworkers reported the synthesis of new chromene, pyrane, and pyranopyridine derivatives bearing the 2-thiobarbituric acid moiety [200]. Initially, they performed the Knoevenagel condensation of 2-thioxo-dihydropyrimidine-4,6(1*H*,5*H*)-dione **280** with aromatic aldehyde 2 in refluxing ethanol and piperidine as the base to render chromeno[2,3-*d*]pyrimidinederivative **281**. 7-Aminopyrano[2,3-*d*]pyrimidine-6-carbonitrile derivative **283** was synthesized in good yield after treatment of **280** and 2-(3,4,5-trimethoxybenzylidene)malononitrile **282** under refluxing ethanol, in the presence of catalytic amounts of piperidine. In a similar manner, reaction with ethyl 3-(4-chlorophenyl)-2-cyanoacrylate **284** afforded ethyl 7-amino-5-(4-chlorophenyl)-4-oxo-2-thioxo-2,3,4,5-tetrahydro-1*H*-pyrano[2,3-*d*]pyrimidine-6-carboxylate **285**. In addition, the reaction of pyrano[2,3-d]pyrimidine **283** with malononitrile **123** in the presence of piperidine afforded pyridopyranopyrimidine **286**. 2-Arylsulfonylamino pyrano[2,3-*d*]pyrimidine **288** was obtained by refluxing compound **283** with benzenesulfonyl chloride **287** in dry benzene. Finally, heating compound **283** with formic acid caused cyclization to give pyrimidopyranopyrimidine derivative **289** (Figure 109). The novel pyrimidines fused at C5 and C6 positions showed good antimicrobial activity.

Pyrrolo[2,3-d]pyrimidines are scaffolds present in a wide variety of compounds with diverse biological activities; however, there are not many methods available to access them. Bhuyan et al. described a microwave-assisted MCR from *N*,*N*-dimethyl-6-aminouracil **290**, aryl glyoxal **291**, and aromatic amines **235** [201]. The reaction took place in AcOH at 110 °C, rendering a new family of 5-arylamino-pyrrolo[2,3-*d*]pyrimidines **292** in excellent yields (Figure 110).

Zhang et al. reported the synthesis of pyrido[4,3-*d*]pyridines by means of an iron-catalyzed vinilogous aldol reaction of Biginelli adducts **293** with aryl aldehydes **2** followed by a base-catalyzed intramolecular aza-Michael reaction [202]. The authors identified that the presence of a methyl group in Biginelli products at position 6, due to the use of acetoacetates normally used in the reaction, could be functionalized by the formation of the corresponding enolate. Fe(III) chloride promoted the vinylogous aldol reaction of Biginelli substrates **293** with aryl aldehydes **2** to render (*E*)-6-arylvinyl-dihydropyrimidin-2(1*H*)-ones **294**; it is important to mention that the presence of the amide moiety in position 6 is necessary to effect this condensation, and the process is very efficient with aromatic substituents while the use of heteroaromatic or aliphatic ones produced a clear drop of the efficiency of the process. Derivatives **294** were further cyclized by means of an intramolecular aza-Michael reaction in the presence of NaOH to furnish pyrido[4,3-*d*]-pyrimidines **295** in good yields (Figure 111).

Verma et al. employed Biginelli adducts **296** as starting materials for the synthesis of imidazopyridines **299** [203]. Thus DHMPs **296**, containing an amide moiety at C5 were heated with pyrane tetraol **297** and chloroacetic acid **298** in refluxing ethanol, rendering final imidazopyridines **299** containing the sugar moiety in good yields (Figure 112); those compounds were subjected to several biological assays, and the authors found that they display antifungal, antibacterial, antioxidant, and anticancer activity.

Guo, Zou, and coworkers developed an enantioselective synthesis of DMHPs **302** with aliphatic aldehydes catalyzed by a chiral phosphoric acid with excellent enantioselectivities [204]. Those compounds were used as starting materials for the total synthesis of natural products Crambescin A and Batzelladine A and several analogues. Chiral DHPMs **4** were treated initially with BuLi and 1,2-dichloroethane to install the pyrrolidine ring. The resulting compounds **300** were then treated with triethyl oxonium tetrafluoroborate to render imidates **301** which in turn were transformed into the corresponding guanidines **302** by heating with ammonium propionate. In this manner, the guanidine core of the natural products was created; those compounds were previously described as advanced intermediates of Crambescin A and Batzelladine A [205] and therefore, this strategy could be considered as a formal synthesis of both natural products (Figure 113).

Bavantula and coworkers described the synthesis of a series of newly fused thiazolo[2,3-b]pyrimidinones bearing a pyrazolylcoumarin moiety [206]. Thiazolo pyrimidinones and pyrazolo coumarins are known as bioactive pharmacophores, and the authors decided to combine them in order to evaluate the potential synergistic influence of both motives. Starting from Biginelli products **303**, they were treated in a three-component reaction with chloroacetic acid **298** and 3-(2-oxo-2*H*-chromen-3-yl)-1-aryl-1*H*-pyrazole-4-carbaldehydes **304** in a mixture of acetic acid and acetic anhydride. After 4–6 h at reflux, excellent yields of polycycles **305** were obtained (Figure 114). Biological studies of these derivatives showed that they display both antibacterial and antitumoral activity.

Nagaraja et al. used an MCR to functionalize Biginelli adducts **4** with a pyrazol moiety [207,208]. To this end, the condensation of DHPMs **4** with diaryl pyrazoles **306** and chloroacetic acid in a mixture of acetic acid/acetic anhydride led the authors to synthesize a new family of pyrazole integrated thiazolo[2,3-*b*]dihydropyrimidinone derivatives **307** in generally good yields (Figure 115); biological evaluation of the final products showed that those compounds display a dual anti-inflammatory and antimicrobial activity.

Both enantiomers of DHPMs typically exhibit different or even opposite biological activities and their asymmetric synthesis is of crucial importance. In this context, Massi and coworkers reported the enantioselective acylation of DHPMs with aromatic enals employing chiral *N*-heterocyclic carbenes (NHC) as the catalyst [209]. To this end, starting racemic Biginelli adducts **4** were treated with enals **308** in the presence of chiral NHC **XIII** and the oxidant **309** at room temperature. Under these conditions, enantioenriched acylated DHMPs **310** were obtained albeit with moderate enantioselectivities (Figure 116).

## 8. Conclusions

Considering the number of DHPM-containing drugs discovered and used in the treatment of multiple diseases, it is not surprising that the amount of newly synthesized DHPMs has exponentially increased in the last two decades. The classical approach to access those scaffolds is the multicomponent Biginelli reaction, discovered more than a century ago. The rebirth in the 1980s of the Biginelli reaction has driven huge synthetic efforts to improve the efficacy of this transformation; however, these efforts have mostly focused on the improvement of the reaction conditions. Additionally, the inherent limitations of the use of three specific components compromise the access of several substitution patterns in the final DHPMs. Mainly in the present century, several variations of these components have been evaluated, together with the modification of the DHPM scaffold itself, which made it possible to synthesize novel Biginelli-like derivatives that possessed new promising biological activities; this review summarized all those efforts, showing the great diversity of new synthetic approaches developed to build different DHPM derivatives.

Regarding the biological relevance of the DHPM substructure, we will witness in the future the appearance of new synthetic methods to increase the synthetic tools already existing to construct DHPMs.

## Data Availability

Not applicable.

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
