# Peer review of "Synthesis of 3,4-Dihydropyrimidin(thio)one Containing Scaffold: Biginelli-like Reactions"

_pharmaceuticals, 2022, doi:10.3390/ph15080948_

Round 1

Reviewer 1 Report

The submitted article “Synthesis of 3,4‐Dihydropyrimidinone Derivatives: Biginelli‐Like Reactions” is a very comprehensive review article. The topic is interesting, due to the biological relevance of the DHPM substructure. The paper is well-written but is rather big which makes it not easy to read. It can be published in Pharmaceuticals after minor revision. To improve the paper the authors should correct certain passages:

·       List of abbreviations used in a paper is missing (especially the abbreviations used above the arrows in the schemes).

·       Page 3 line 82 - „In contrast, Biginelli‐like reactions, were one of more components of the multicomponent process were modified, were only laterally treated”. – sentence unclear

·       Page 4 line 114 - “…functionalized a‐benzylidene b‐keto esters 3, This derivatives were synthesized…” – after “3” should be comma or dot? In the same line later “…were synthesized in order to to establish structural and conformational determinants in calcium channel modulation…” – two times “to”.

·       Please revised “et al” (e.g. on page 9 et al. and et al) and “in vitro” (eg on page 10 in vitro or in vitro -  once written in italics and once without italics), please unify in a whole paper

·       Page 10 line 285 - NH4VO3 “as catalysis” or “as a catalyst”? please check

·       Page 12 line 339 - “of different of different catalysts”

·       Scheme 28 – number 1 for urea/thiourea is missing

·       Page 17 line 468 – for Chikhale et al. the reference is missing

·       Page 21 line 566 - “In the case of using cyclohexanone 6 as the enolizable ketone” it should be 86 instead 6

·       Scheme 43 – “Scheme 43. Biginellilike condensations of 6methoxy1tetralone catalyzed by acidic IL VIIII” - should be VIII instead VIIII

·       In the schemes once in MW and once is μwaves an once is μ-waves

·       Page 27 line 697 developed by Shah et al. – see reference 126, you have Shan, please change it for Shah

·       Scheme 64 – please upload new figure – 152 is barely visible

·       Scheme 77 – please check is it correct (long space between Et and Me after R2, under R1=R2 is Bn? Is it ok?)

·       Page 39 line 981 - “…were obtained by reaction with with ethyl 2‐cyano‐2‐[4‐(substituted)phenyldiazenyl]acetates…” two times “with”.

·       See reference 177, it should be König instead Köning (you should check spelling in a whole references)

·       Page 43 line 1068 - “…in refluxing methanol (Scheme 13).” It should be Scheme 90?

·       Page 50 line 1224 – “antigungal properties” should be antifungal properties

Reviewer 2 Report

The article Synthesis of 3,4‐Dihydropyrimidinone Derivatives: Biginelli Like Reactions after consideration of major comments.

The review is imperative and displayed the recent methods for synthesis of 3,4‐Dihydropyrimidinone derivatives

.

1)      Titled should be reflect the review contents so it is better to change into

Synthesis of 3,4‐Dihydropyrimidin(thio)one containing scaffold : Biginelli‐ Like Reactions

2)      Abstract should be improved to be more informative

3)      Scheme 2, authors should indicate how compound 6 converted to

4)      Scheme, authors should explain why phenacyl bromide reacted at N 3 not N1.

5)      Scheme 6, which substitute give which product and why.

6)      In vitro or in vivo should be italic (line 166).

7)      Scheme 8, the product is not dihydropyrimidone  (24) it is  selenium derivatives .

8)      Scheme 9, the product is dihydropyrimidnone isostere.

9)      Line 188., I ask if PW is abbreviation for tungstophosphoric acid?

10)   Scheme 10 authors should mention the effect of different catalyst.

11)   Scheme 14,15 , 18 etc mention synthesis of fused ring system not dihydropyrimidinones?

12)   It is better to summarized Biginelli reactions in table to compare different methods regarding conditions and reactants .

13)   It is better to correlate structures to corresponding biotical activity.

Round 2

Reviewer 2 Report

the review is accepted in the present form